# Intersection of the Orphan G Protein-Coupled Receptor, GPR19, with the Aging Process

**DOI:** 10.3390/ijms232113598

**Published:** 2022-11-06

**Authors:** Stuart Maudsley, Deborah Walter, Claudia Schrauwen, Nore Van Loon, İrem Harputluoğlu, Julia Lenaerts, Patricia McDonald

**Affiliations:** 1Receptor Biology Lab, University of Antwerp, 2610 Antwerpen, Belgium; 2Moffitt Cancer Center, Tampa, FL 33612, USA

**Keywords:** GPR19, GPCR, aging, therapeutics, longevity, stress, resistance

## Abstract

G protein-coupled receptors (GPCRs) represent one of the most functionally diverse classes of transmembrane proteins. GPCRs and their associated signaling systems have been linked to nearly every physiological process. They also constitute nearly 40% of the current pharmacopeia as direct targets of remedial therapies. Hence, their place as a functional nexus in the interface between physiological and pathophysiological processes suggests that GPCRs may play a central role in the generation of nearly all types of human disease. Perhaps one mechanism through which GPCRs can mediate this pivotal function is through the control of the molecular aging process. It is now appreciated that, indeed, many human disorders/diseases are induced by GPCR signaling processes linked to pathological aging. Here we discuss one such novel member of the GPCR family, GPR19, that may represent an important new target for novel remedial strategies for the aging process. The molecular signaling pathways (metabolic control, circadian rhythm regulation and stress responsiveness) associated with this recently characterized receptor suggest an important role in aging-related disease etiology.

## 1. Introduction

GPCRs constitute one of the most diverse groups of transmembrane signal transducers that control a panoply of physiological processes in many species, ranging from *C. elegans* to *Homo sapiens*. The GPCR transmembrane superfamily is characterized by a common seven α-helical transmembrane domain motifs. GPCRs represent one of the most therapeutically important molecular targets in clinical medicine [1,2,3,4]. GPCRs facilitate communication between cells in tissues across long distances in the body, thereby enabling the capacity for systems-level therapeutic actions [5,6,7,8]. In addition to this long-distance signal transduction role, GPCRs also regulate intracellular signal transduction scenarios that regulate cellular stress responses [9]. Underlining their importance to therapeutic development, medicines have been historically developed to exploit these GPCR systems for many years, even before the discovery of GPCRs themselves [10]. Our research, as well as others, have begun to demonstrate that the GPCR systems can be targeted to control multiple physiological systems across the body and thus present an ability for GPCR ligands to control complex disorders such as pathological aging [2,5]. Implicit in this systems-wide functionality is the connected concept that GPCR activity is both vital for long-range tissue-to-tissue communication [5] and also the creation of stress-sensitive GPCR signaling networks at the single cellular level [9]. In this regard, we will highlight the Class A orphan GPCR, G protein-coupled receptor 19 (GPR19), as a potential novel regulator in the metabolic aging process. Here we will outline how generic GPCR signaling systems, as well as those specific to GPR19 activity, show a strong functional intersection with the aging process and thus represent potential novel targets for aging-related disease treatment [2].

### 1.1. GPCR Signaling

Given the importance of GPCR signaling in controlling physiological functions, multiple investigations over several decades have sought to fully appreciate how these transmembrane receptors control cellular activity at the signal transduction level. From a functional receptor pharmacological approach, controllers of these receptors were originally designed to exert either a simple positive effect (increasing the activity of downstream signaling systems, e.g., adenylate cyclase) or by inhibiting this activity by occupying the receptor and antagonizing the positive actions of stimulatory ligands. Therapeutic agents were classified as simple agonists (stimulatory) or antagonists (inhibitory) based on the concept that receptors could exist predominantly in two distinct states, i.e., inactive and active. Over the next thirty years, intense research largely confirmed this ‘two-state’ GPCR model [11,12,13,14,15,16,17]. Using specific site-directed mutagenesis of key residues in GPCRs [18], it was demonstrated that GPCRs indeed likely exist in a spontaneous equilibrium between two conformations, i.e., active (R*) and inactive (R). The active conformation is naturally stabilized by agonist binding or, in these numerous experiments, by residue mutation that serves to relax intramolecular constraints [18,19,20,21,22,23]. In this initial functional model, GPCRs transmit signals through their capacity to act as guanine nucleotide exchange factors for heterotrimeric guanine nucleotide-binding proteins (G proteins) in response to stimulatory ligand binding (or via constitutively activating point mutagenesis). Ligand-mediated G protein activation is initiated through conformational rearrangement of the heptahelical GPCR core and juxtamembrane loop regions, eventually catalyzing the exchange of GDP for GTP on the receptor-associated Gα subunit [24,25,26,27,28]. Guanine nucleotide exchange (GDP for GTP) then initiates the dissociation of the heterotrimeric G protein from the GPCR, followed by the dissociation of the G protein heterotrimer releasing free GTP-bound α and βγ subcomplexes. These two signaling components can stimulate, inhibit or physically recruit multiple downstream signal transduction effectors, e.g., adenylyl cyclase (AC), phospholipase C (PLC), GPCR kinases (GRKs) or GRK-interacting proteins [29]. In this manner, the heterotrimeric G protein can transmit information to the intracellular milieu about the qualitative and quantitative nature of specific extracellular stimuli [30,31].

There are 16 Gα, 5 Gβ and 13 Gγ subunits in humans, which allows many different aggregations and signaling outputs [32]. The responses range from activation to inhibition, and G proteins are classified based on the downstream function of the alpha (α) subunit into four families, Gαs, Gαi/o, Gαq/11, and Gα12/13. Gαs activates adenylyl cyclase, which then converts ATP into the second messenger cyclic adenosine monophosphate (cAMP) and activates the cAMP-dependent pathway. Conversely, Gαi/o decreases cAMP levels [33]. Apart from the classical G protein signaling, multiple lines of research have validated the existence of multiple non-G protein signaling activities, such as β-arrestin [34,35,36], proto-oncogene tyrosine-protein kinase Src (c-Src) [37,38,39] and the ADP-ribosylation factor GTPase activating protein-2 (GIT2) [40,41,42]. Thus, it is now clear that GPCR signaling is more nuanced and complex than previously thought. The dogma of simple physiological G protein signaling specificity of downstream signaling was broken with the initial demonstration that alternative modes of signaling, e.g., the first being the β-arrestin paradigm, are physiologically relevant and are therapeutically tractable [28,34,35,36,43,44,45,46,47]. β-arrestins were first characterized as negative regulatory proteins for signaling through G proteins and were considered responsible for GPCR internalization and separation from G protein engagement [48,49,50]. β-arrestin has been subsequently shown to serve as a scaffold for a variety of signaling complexes associated with GPCR signaling pathways [28,34,43].

While the field of non-G protein-dependent GPCR signaling has been historically dominated by β-arrestin activity, several other modalities have been demonstrated, including Janus kinase 2 (JAK2) [51], 14-3-3 proteins [52], RGS proteins [53], Proline-rich tyrosine kinase 2 (PYK2) [39,54] and the ADP-ribosylation factor GTPase-activating protein 2 (GIT2) [40,55]. With regards to this last GPCR signaling adaptor, GIT2, several strong contrasts to β-arrestin signaling have been reported, thus making this paradigm an interesting one to compare with the pioneering β-arrestin pathway [2,9,29]. With the specific link to aging paradigms, it has been shown that β-arrestin activation leads to increased DNA damage in stress conditions, degradation of p53, suppression of NF-kB and the promotion of apoptosis [56,57]. Conversely, it appears that GIT2 may represent a natural mechanism to prevent aging-associated molecular and cellular damage. The GPCR-kinase interacting protein (GIT) family of proteins (GIT1 and GIT2) were originally identified as GRK and GPCR interacting proteins [58]. Subsequently, it has been shown that especially GIT2 exerts systemic effects upon a multitude of signaling and physiological systems, including oxidative stress resistance [59], glucose metabolism [60], circadian rhythm regulation [61], mitochondrial activity [41,60], DNA damage repair response [40,62], immunosenescence [61] and gender-specific lifespan regulation [41]. Given this information, it is unsurprising that additional interest in non-G protein-dominated GPCR signaling pathways has been shown with respect to the proposal that signaling paradigms such as β-arrestin and GIT2 may indeed possess specific benefits for the GPCR-mediated interdiction of aging-related disease [35,40,42,63,64,65,66,67]. For both signaling paradigms, significant evidence has shown that these two GPCR signaling modalities converge on the regulation of energy metabolism and DNA damage/repair [34,42,56,57,59,60,61,62,66,68,69,70,71,72,73,74,75].

### 1.2. Aging and GPCR Functionality

Aging and age-related damage of cellular proteins and nucleic acids are inevitable results of lifelong cellular metabolic activity [40,60,62,76,77,78]. This cellular damage occurs most frequently because of the production of deleterious metabolites, e.g., reactive oxygen species (ROS), as by-products of energy management processes such as mitochondrial oxidative phosphorylation [9,79,80,81]. There are many other sources of aging-related damage, but many lines of evidence have suggested that this process is one of the most potent sources of recurring cellular damage and, ultimately, age-related disease [82,83,84,85,86,87]. This stress-related damage essentially degrades the functionality of active signaling systems as well as reactive cytoprotective cellular systems that exist to combat the metabolically induced cellular damage [2,9,88,89,90,91]. In recent years it has been demonstrated that—as with many other forms of cellular and tissue signaling [90,92,93]—stress response and DNA damage repair processes are strongly controlled and regulated by signaling networks composed of multiple GPCR types [5,9,29,63,64,65,66,73,94,95]. Thus, well-informed therapeutic targeting of GPCRs holds a strong promise for the generation of a broad series of anti-aging therapeutics.

GPCRs represent one of the most important therapeutic targets for controlling disease generation and progression [96,97]. Underlying their importance in the broad range of biological functions, GPCRs are the most structurally diverse family of transmembrane proteins. The superfamily comprises more than 800 proteins, which are grouped based on evolutionary homology and common physiological ligands. Human GPCRs are divided into six major classes, class A (rhodopsin-like), class B1 (secretin receptor-like), class B2 (adhesion receptors), class C (metabotropic glutamate receptor-like) and class F (frizzled-like) subfamilies, as well as the taste 2 sensory receptor subfamily [98]. GPCRs sense a tremendous variety of stimulating entities ranging from photons, ions, and neurotransmitters, to complex hormones and exogenous animal toxins. This nuanced sensory system allows cells to react quickly to diverse endogenous and environmental perturbations [99]. GPCRs are of major interest for drug development due to their regulatory function for a multitude of physiological processes, as well as their accessibility for exogenous ligands. Hauser et al. evaluated in 2017 that 475 FDA-approved drugs target GPCRs, which is 34% of all FDA-approved drugs [96].

### 1.3. GPCR-Based Control of Aging-Related Mechanisms

As we have previously described, it has been proposed that non-G protein-dependent signaling paradigms may hold specific promise for the amelioration of aging-associated diseases [2,29,42]. Arrestin-dependent signaling is one of the most important and well-characterized of these signaling modalities [34,94,100]. Recent translational research has demonstrated that the β-arrestin-based signaling modality can generate a clinically relevant signaling paradigm [36,46,47,101]. Using a novel, in vivo-biased signaling demonstration, it was found that β-arrestin-GPCR complexes likely possess the ability to elicit a coherently conserved signaling cascade across multiple tissues, distinct from the G protein paradigm, even after a month of continuous drug dosing of the biased agent [46]. Furthermore, and in stark contrast to G protein-sourced signaling that primarily controls acute intermediary cell metabolism events (e.g., intracellular calcium mobilization), β-arrestin-dependent signaling generates a strong transcriptional and translational signaling functionality [102,103,104,105]. This aspect of β-arrestin-dependent signaling, therefore, lends itself to the concept that β-arrestin-biased ligands could be rationally designed to therapeutically regulate complex protein networks that underpin many complex aging-related diseases, e.g., Type II Diabetes Mellitus (T2DM), neurodegeneration and cancer [29,91,106]. The expression levels and signaling activity of β-arrestin have been shown to be involved with metabolic aging conditions such as Alzheimer’s disease [107,108], Parkinson’s’ disease [109,110,111], T2DM [112,113,114], osteoporosis [44,46,115] and schizophrenia [116,117]. In a similar vein, research has also demonstrated that the GPCR adaptor GIT2 can play a pivotal role in metabolic aging conditions and disorders, including neurodegenerative diseases [118,119], T2DM [60] osteoporosis [120,121] and psycho-affective disorders [122,123]. In addition to these two major forms of GPCR signaling adaptors, considerable evidence has been generated to demonstrate the role of GRKs, PYK2 and JAK2 in aging-related conditions associated with metabolic dysfunction [2]. This consistent finding, therefore, suggests that perhaps the association of GPCRs with these non-G protein signal adaptors may create a stress-sensory mechanistic network of receptors that naturally control the severity of these conditions [9]. Hence, we contend that the molecular intersection between cellular damage control and metabolic dysfunction systems plays a pivotal role in regulating the balance of energy regulation and cellular stress responses. While presenting tremendous promise for the future of pharmacotherapy via precision medicine, the multiplicity and importance of β-arrestin and GIT2 in a plethora of physiological processes does raise the possibility of incurring systemic side effects resulting from excessive, or even total β-arrestin or GIT2 bias, at the expense of G protein activity. Hence, an intelligently informed and subtle approach to signaling bias exploitation should be taken. Potentially the development of novel biased ligands targeting the β-arrestin-or GIT2-associated receptorsome currently represents a vital new pharmacotherapeutic domain [40,44,63,124,125].

## 2. GPR19 and Aging-Related Activity

GPR19 was first identified by O’Dowd et al. [126]. A sequence analysis of the cloned cDNA predicted that the 415-amino acid long protein contains seven transmembrane repeats that characterize the GPCR superfamily [126]. The highest expression levels of GPR19 are found in the brain, testis, and lymph nodes [127]. The GPR19 gene was assigned to the chromosome position 12p13.2-p12.3, approximately 40kb from the CDKN1B gene [126,128]. It was further observed that the physical mapping of GPR19 on the chromosome was frequently rearranged in cancer cells [128].

GPR19 is still officially classified as an orphan class A GPCR by IUHAR/BPS [129]. While GPR19 has been associated with the ligand adropin in multiple studies [130,131,132] a direct interaction could not be confirmed in a deorphanization study by Foster et al. [133]. Using phylogenetic sequence alignment and clustering of over 300 Class A GPCRs GPR19 was demonstrated to co-cluster loosely with oxytocin, vasopressin, and gonadotropin-releasing hormone receptors [134]. Within this cluster, GPR19 was more closely related to GPR154 (Q6JSL8, Q6JSL4) which has subsequently been identified as a receptor for Neuropeptide S [135].

GPR19 is likely Gαi-coupled and, therefore, potentially linked to the attenuation of adenylyl cyclase activity [130,136]. The constitutive activity of some GPCRs allows the screening of even orphan receptors by simply overexpressing them and detecting the resultant alterations in prevailing cAMP or Ca^2+^ levels [136]. In a study by Rao et al. [130], it was shown that GPR19 and its activation by a proposed cognate ligand for this receptor, i.e., the peptide adropin, upregulate phosphorylated ERK and E-cadherin expression. This effect was already observed in cells overexpressing GPR19, suggesting some constitutive activity and was further enhanced when adropin was added [130]. However, when constitutive activity was studied in GPR19 and at least 30 other orphan Class A receptors, no constitutive activity was observed. This specific study focused on cAMP-dependent mechanisms and defined activating G protein activity at a 200% elevation of cAMP levels and inhibiting activity at a decline of 40% from the baseline signal [137]. These results may suggest that Gαi and Gαs protein-induced cAMP signaling is probably not one of the most prevalent signaling pathways induced by GPR19 in the absence of a stimulating ligand. Further studies have subsequently observed constitutive recruitment of β-arrestin in GPR19, and all other studied orphan GPCRs (82 in total, including GPR19) and suggested that this is a common feature of most GPCRs [138].

GPR19 signaling has been associated with multiple activities linked to dysfunctional aging, e.g., cell cycle control, cancer metastasis, adipocyte proliferation and lipid metabolism, stress-associated apoptosis, mitochondrial function, and diabetes [130,132,139,140]. Moreover, the putative ligand adropin has shown multiple links to energy metabolism, cell proliferation and cancer. Indeed, adropin levels in the brain have been shown to decrease with advanced age, and this diminution has been linked to increased oxidative stress in neuronal tissues [141]. In addition to this, adropin levels have more recently been demonstrated to also correlate with aging-related neuropathology in humans [142]. In the following section, we will outline in more detail how GPR19 and adropin were shown to be involved in the different molecular signatures of aging.

Recently it has been demonstrated that an exemplar GPCR and adaptor protein combination, i.e., the human RXFP3 receptor and GIT2, holds tremendous promise for the future GPCR-based control of aging trajectories [40]. In this context, the stimulation of the RXFP3 has been shown to prevent or even repair existing DNA damage. Hence, GPCR-stimulating ligands that promote GIT2-based signaling can potentially stall the accumulation of unrepaired DNA damage that can drive the aging process [42,62]. In addition to RXFP3, recent evidence has suggested that further GPCR signaling systems can also be effectively targeted to control aging-associated pathological activity. In this regard, due to its close link to aging-related mechanisms and deficiencies in energy metabolism, the orphan receptor GPR19 [126] might be a suitable drug target for such longevity-controlling therapeutics. Even though GPCRs have been studied for decades, their entrained signaling paradigms and the potential for ligand-directed bias between them [47,101] is still not fully understood. While the GPCR superfamily is the most intensively studied drug target family [3], only a small minority (approximately 46) is exploited therapeutically. In addition to this relatively poor exploitation of this tremendous therapeutic resource, currently, 121 GPCRs are still considered orphan receptors and thus represent yet more underappreciated molecular targets. Given the high potential of these undiscovered drug targets, further research to understand the molecular function of orphan receptors is needed. It has furthermore recently been hypothesized that GPCRs may employ multidimensional signaling pathways [94,101]. Understanding the preferences of receptors and ligands to induce specific signaling pathways over others is applicable to optimize therapeutic development.

### 2.1. GPR19 and Energy Metabolism

Adropin is a peptide encoded by the energy homeostasis-associated gene (ENHO) and was first described in 2008 by Kumar et al. [143]. Since then, it has been identified as a regulator of glucose homeostasis and lipid metabolism [131,132,144,145]. In mice, adropin-overexpression attenuated body weight gain on the high-fat diet and adropin treatment had beneficial effects in regulating energy homeostasis. In obese mice on a high-fat diet, adropin treatment reduced fasting blood glucose and increased glucose tolerance, indicating a potential increase in whole-body insulin sensitivity. In this model, adropin treatment also reduced hepatic glucose production and simultaneously improved hepatic insulin sensitivity [144]. Additionally, human studies confirmed the role of adropin in energy metabolism. Plasma adropin concentrations have been demonstrated to be sensitive to dietary macronutrients and have been shown to increase with dietary fat content [146]. It was also shown that serum adropin levels were significantly lower in diabetic patients [147,148]. These results suggest that adropin treatment can attenuate metabolic abnormalities in obesity as well as T2DM [140]. Thapa et al. [132] demonstrated how GPR19 likely plays a role in metabolism by regulating mitochondrial respiration. Adropin induces mitochondrial fuel substrate utilization towards using more glucose. In cardiac cells, it was shown that adropin-induced GPR19-MAPK-PDK4 signaling regulates pyruvate dehydrogenase (PDH), a rate-limiting enzyme in glucose oxidation. Thereby, increased adropin levels decrease inhibitory PDH phosphorylation, which ultimately leads to increased mitochondrial respiration and O_2_ consumption [132]. Apart from downregulating PDK4, adropin was shown to regulate the activity of PGC-1α in muscle cells, which is a key transcriptional regulator of oxidative metabolism. It was suggested that adropin increases PGC-1α acetylation by inhibiting the PGC-1α deacetylase Sirtuin-1 (SIRT1). Increased acetylation of PGC-1α usually inhibits the activity of transcriptional factors, such as PPARs, which controls the expression of various genes, including Cpt1b and Pdk4 [145].

### 2.2. GPR19 and Cell Cycle Regulation

The cell cycle and proliferation regulation are important factors in aging and cancer. Dysfunction can induce apoptosis and facilitate cancer propagation but can also cause senescence-related inflammatory damage. GPR19 facilitates breast cancer cell metastasis by contributing to the promotion of mesenchymal to epithelial transition [130]. Overexpression, or ligand (adropin) activation, of GPR19, was shown to induce mesenchymal-like breast cancer cells to adopt an epithelial-like phenotype. This activity was linked to a responsive stimulation of the ERK/MAPK pathway and the resultant elevation of E-cadherin expression. The control of epithelial characteristics at secondary tumor sites is now understood to be an essential step in the tumor colonization process. In this context, it was proposed that GPR19 may be involved in metastasis by promoting the mesenchymal-epithelial transition (MET) through the ERK/MAPK pathway and thus augment the potential colonization of metastatic breast tumor cells. In lung cancer cells, the expression of GPR19 has a potential supporting role in G2-M cell cycle progression [149], and in metastatic melanoma, an increased expression of GPR19 was observed [150]. GPR19 signals through the MAPK and ERK1/2 pathway to the DNA in the nucleus, which can induce changes, such as starting cell division [130]. ERK1/2 activity is required for G0–G1 cell cycle transition and the passage of cells through mitosis or meiosis [151,152]. Both overexpression of GPR19 and adropin can induce activation of ERK and protein kinase B (AKT) signaling pathways, which are involved in controlling pre-adipocyte proliferation and differentiation [153,154]. Under the influence of cellular stimuli, including adropin, phosphatidylinositol 3-kinase (PI3K) induces the phosphorylation and activation of (AKT) [155]. Phosphorylated AKT triggers mTOR and attenuates apoptotic factors such as glycogen synthase kinase 3 β (GSK3β) [156]. The PI3K/AKT/mTOR signaling pathway regulates a wide variety of cellular functions, including survival, proliferation, growth, metabolism, angiogenesis, and metastasis [157]. The PI3K pathway is also associated with insulinotropic activation of anabolic metabolisms, including glycogen and lipid synthesis [158]. Defective AKT signaling is associated with neurodegenerative diseases such as Alzheimer’s, Parkinson’s, and Huntington’s diseases. Moreover, dysregulation in the AKT/GSK3β signaling pathway is linked to the neuropsychiatric diseases schizophrenia and bipolar disorder [156]. Moreover, serum adropin plays a role in endothelial function and protects endothelial cells from tumor necrosis factor-α induced apoptosis via AKT, ERK1/2 and eNOS kinases [155].

### 2.3. GPR19 and Oxidative Stress

Adropin was demonstrated to be an independent predictor of coronary atherosclerosis in diabetic and non-diabetic patients [148]. Similar trends showing lower adropin serum levels than controls were observed in patients with hypertension, atrial fibrillation, and many other cardiovascular diseases [159,160]. Linked to the several associations of adropin to cardiovascular diseases, it was shown that GPR19 is increased after myocardial infarction and leads to increased ischemia-reperfusion injury, oxidative stress, and apoptosis [139]. This suggests that GPR19 might be an important regulating factor of oxidative stress. In line with these results, our lab and Williams et al. [161] both demonstrated that GPR19 was upregulated in a different model of oxidative stress. Our lab used the GIT2 KO murine model of aging. GIT2 has been demonstrated to be a key regulator of aging mechanisms, many of which are linked to oxidative stress [42]. Williams et al. [161] showed that GPR19 was upregulated in a nuclear factor erythroid 2 (Nfe2) KO Danio Rerio model. Nfe2 is a cap’n’collar basic leucine zipper transcription factor involved in the oxidative stress response. Nfe2 regulates transcription by binding cis-antioxidant response elements (cis-AREs). Several of these cis-AREs were found on the GPR19 gene and other upregulated proteins in the Nfe2 KO zebrafish model. Therefore, Nfe2 could act as a suppressor of GPR19 [161]. A further link to aging is the upregulation of GPR19 when testing adaptogens [162]. Adaptogens are natural compounds or plant extracts that can augment cellular stress adaptability and thus enhance organism survival in times of stress. Given that molecular aging is driven by stress-induced (e.g., oxidative stress) protein/lipid/nucleotide damage, it is clear that adaptogen employment may be able to reduce this stress-induced pathology and thus reduce the rate of age-associated damage accumulation. Using human glioblastoma neuronal cells, an adaptogen-induced potentiation of GPR19 expression was found using unbiased RNA sequencing analyses. This work suggests that human cells can be induced to demonstrate a better stress-response capacity that potentially includes elevated GPR19 signaling activity [162].

### 2.4. Circadian Rhythms

Recently it was shown that GPR19 plays a crucial role in the modulation of the circadian clock in the suprachiasmatic nucleus (SNC) [163]. GPR19 KO murine models presented a delayed onset of circadian locomotor activity. The circadian period was prolonged in GPR19 KO murine models, compared to the WT, when the animals were kept in constant darkness. This study further demonstrated that GPR19 possesses a cAMP-responsive element (CRE) motif in the promoter region, which makes GPR19 expression dependent on cAMP levels. Deletion of this region disrupts the cycle-dependent expression of GPR19 in the suprachiasmatic nucleus, which is highest during the daytime and lowest during the nighttime. The deficiency of GPR19 mainly downregulated rhythmic genes that peaked during the nighttime. GPR19 was also shown to aid an effective phase shift after a light pulse.

## 3. Functional GPR19 Molecular Signatures

Using publicly available data, e.g., the Gene Expression Omnibus (GEO: [164]), a considerable degree of inference can be made with respect to the functional activity of a specific protein [91,106]. In this respect, we have performed a co-expression response analysis to identify a functional family of proteins associated with GPR19 signaling activity. Using GEO, a query for “GPR19” resulted in 4336 dataset results mentioning GPR19. For datasets where a different gene expression of GPR19 was apparent in the different conditions, a GEO2R analysis was performed and differentially expressed (DE) genes were extracted when GPR19 was also among the significantly differentially expressed proteins (*p*-value = 0.05). Identifying GEO profile results in which a profound and consistent alteration in the expression of GPR19 was observed, we then prioritized 7 datasets where GPR19 expression was increased (GSE8157, GSE31102, GSE49506, GSE10309, GSE12881, GSE31812, GSE9754) and 9 datasets where GPR19 expression was decreased (GSE14428, GSE16048, GSE37894, GSE47363, GSE28598, GSE14773, GSE49185, GSE5668, GSE23031). The normalized GEO datasets were further filtered using R. To filter out DE genes linked to altered GPR19 expression, all genes significantly altered were compared between the different datasets with the BioConducter package vendetail (https://bioconductor.org/, accessed on 1 October 2022). The data was visualized, and lists of 1000 genes were extracted in a ranked order with genes present and common to the highest number of GSE datasets at the top of the matrix (Appendix A—Upregulated; Appendix A—Downregulated). For the upregulated GPR19 data cohort, one factor (GPR19 itself) was upregulated across all GSE datasets, and 12 factors were common across these seven GSE datasets (FASN, FLCN, H6PD, TMTC4, FKBP11, ACLY, NAMPT, TES, CAB39L, STRBP, DIXDC1, DDX21), 113 factors were common to 5 different GSE datasets, 595 factors were common to 4 different GSE datasets with 279 factors common to 3 different GSE datasets (Figure 1A). With respect to the 9 datasets in which GPR19 was consistently downregulated, we found only GPR19 itself was consistent across all nice datasets. However, 11 factors were commonly regulated across eight GSE datasets (AURKB, CTSB, FSCN1, GART, GSG2, PLPP3, TMPO, SMIM19, FAM102A, LAMP2, ITGB3BP), 127 factors were common to seven GSE datasets, 641 factors were common to six GSE datasets, and 220 factors were common to five GSE datasets (Figure 1B).

Inspecting the correlation of protein expression, either with GPR19 up- or downregulation, at the highest level, a strong functional intersection between GPR19 activity and aging was apparent. Hence many of the most closely-correlated factors associated with GPR19 upregulation are tightly linked with metabolic aging (FASN: [165]; ACLY [166]; NAMPT [167]; CAB39L [168]), mitochondrial and antioxidant activities (H6PD [169]; DIXDC1 [170]), damage-related cell cycle alterations/cancer (FLCN [171]; TES [172]), alterations in unfolded protein management linked to metabolic imbalances (FKBP11 [173]) and oncogenesis (TMTC4 [174]), DNA damage (DDX21 [175], and body weight (STRBP [176]. Performing a similar investigation of the most closely associated factors in GPR19 downregulation paradigms associations with DNA damage/cell cycle control (AURKB [177]; TMPO [178]; ITGB3BP [179]), metabolic aging (CTSB [180]; GART [181]; PLPP3 [182]; FAM102A [183]), inflammation (FSCN1 [184]), mitochondrially associated autophagic activity (LAMP2 [185]), cell growth and oncogenesis (GSG2 [186]), and aging-related dementia (SMIM19 [187]) were observed.

To evaluate the functional signaling activities of these GPR19 signatures, we next performed a hypergeometric overexpression analysis (ORA) WikiPathways enrichment investigation (ensuring at least n = 2 for specific protein involvement in a pathway and an enrichment probability of <0.05) on the different levels of GSE factor dataset commonality of upregulated or downregulated GPR19-associated factors (Up—analysis of factors common to at least 6 datasets, to at least 5 datasets, to at least 4 datasets and at least 3 datasets: Down—analysis of factors common to at least 8 datasets, to at least 7 datasets, to at least 6 datasets and at least 5 datasets). Collating these pathway analyses, we were able to identify pathways that were consistently and significantly represented at different levels of numerical dataset size investigation (Figure 2). We next assembled clusters of significantly enriched WikiPathways that were found using at least three out of the four levels of dataset size investigation (Figure 3). These were then directly compared between the Up or Downregulated GPR19 input data with respect to the quantitative and qualitative nature of the pathways (i.e., type of pathway activity and the number of closely associated pathways). Here it was evident that the upregulated GPR19 data cohorts were more strongly associated with ‘DNA damage response’, ‘Cholesterol management’, ‘Stem Cell management’ and the ‘Unfolded Protein Response (UPR)’. Conversely, the downregulated GPR19 data cohorts were more strongly associated with ‘Interleukin signaling’, ‘mTOR signaling’, ‘IGF-1 signaling’ and ‘Nuclear Laminopathies’. All of these diverse functions strongly converge with respect to the pathomechanisms linked to pathological aging (DNA damage—[73]; Cholesterol regulation [188,189,190]; stem cell management [191,192]; UPR [193,194]; mTOR signaling [195,196]; IGF-1 signaling [197,198]; laminopathies [199,200]).

While the pathophysiology of aging can be described via a convergence of multiple signaling mechanisms, an orthogonal mechanism of defining aging can be achieved using latent semantic analyses of millions of peer-reviewed documents available at PMC/NCBI to create a user-defined molecular signature of aging [201,202,203]. Using multiple latent semantic analyzer applications (GLAD4U [204], PubPular [205], Geneshot [206]), a user-defined descriptive list of proteins was defined. This final ‘aging-specific’ dataset (227 proteins) was formed using proteins common to at least 2/3 of the different analyzer result streams. To functionally benchmark this aging dataset, an enrichment analysis, i.e., MSigDB MSigDB_V7_C5_GO_Biological_Processes was performed using GeneTrail v3.2 (https://genetrail.bioinf.uni-sb.de/, accessed on 1 October 2022). The results of this enrichment analysis are indicated in Appendix A and Figure 4A. In this analysis, the GO group Aging (GO:0007568) demonstrated the highest degree of enrichment probability (*p* = 1.05 × 10^−35^) compared to the other enriched categories. This benchmarking therefore demonstrates that this dataset, created in an unbiased manner, is indeed a valid representative of the Aging process. Using this dataset, we next interrogated which were the most aging-specific components of the combined GPR19 Up and Downregulated factors (Figure 4B) through the identification of the intersection between these two sets. Here we found 28 factors that may represent perhaps the most important proteins that could mediate the age-controlling aspects of GPR19 biology. These proteins included multiple factors shown to exert a strong regulatory role in the aging process, e.g., DNA damage repair factors BLM [207], XRCC5 [208], RECQL4 [209], FAAP100 [210], FOXM1 [211], NUCKS1 [212], FEN1 [213]; energy metabolism factors UCP2 [214], PARP1 [215], HMOX1 [216], ARG2 [217], IDE [218]; cell cycle/fate control factors CDKN1A [219], CDK5 [220], WWC1 [221], CENPW [222]; transport and proteostasis regulation factors ZMPSTE24 [223], HSPA9, also known as Mortalin [224], SQSTM1 [225], PPM1L [226], PICALM [227]. To assess the degree of specificity of this cohort of GPR19-Aging-associated factors, we tested multiple (n = 10) random datasets the same numerical size as the unbiased aging dataset to assess the potential for random intersections between the GPR19-associated dataset with one the same size as the aging-specific dataset. Using multiple assessments of the degree of random overlap, a significantly lower level of random data overlap was obtained from the results of ten random assessments using randomly generated protein datasets (https://molbiotools.com/randomgenesetgenerator.php, accessed on 1 October 2022). Thus, the overlap we observed is both functionally relevant to the aging process and is also significantly different from the level of GPR19-Aging data overlap that occurs at random (Figure 4C). We next subjected this GPR19-Aging subset (28 proteins) to both functional interaction and cluster analysis (k-means clustering) using STRING (https://string-db.org/, accessed on 1 October 2022). With the use of similar k-means clustering algorithms we have used previously [228], we demonstrated that the 28-protein GPR19-Aging cluster could be rationally separated into four main functional clusters (Figure 5). This algorithmic clustering resulted in the creation of rational protein groups (based on molecular and text-based interaction analysis) associated with (i) mitochondrial function and energy management, (ii) proteostatic protein stability management, (iii) DNA damage regulation and (iv) cell death/fate processes. Taken together, these clusters represent a clear view of the potential roles of GPR19 in the aging process. Rather than limiting ourselves to one form of algorithmic clustering, we next used a further protein-protein-interaction (PPI) methodology to potentially screen for further molecular interactions associated with the GPR19-Aging paradigm. We deployed the PPI enrichment suite of NetworkAnalyst (https://www.networkanalyst.ca/, accessed on 1 October 2022) using the specific IMEx Consortium (https://www.imexconsortium.org/, accessed on 1 October 2022) database as a standard platform. Using the input of 28 proteins, a Minimum Order network was created based on this network (Figure 6A). This expanded network thus included proteins that may bridge multiple factors within the original 28 protein input set and therefore provide a richer appreciation of the functionality at a global level of the GPR19-Aging set. One of the hallmark activities of GPCRs in the aging process appears to be their ability to sense and regulate alterations in cellular status to prepare for detrimental periods of reduced energy or protection from oxygen radicals [2,9,29]. It is interesting to note that within the expanded network, the most significantly populated GO biological process term was ‘Cellular Response to Stress—GO:0033554′ with an enrichment *p*-value of 2.1 × 10^−36^ (Figure 6B). The protein factors that constituted this response (SQSTM1, CDKN1A, BLM, PARP1, PSEN1, XRCC5, FOXM1, HMOX1, FEN1, FAAP100) were interestingly drawn from across many of the functional k-means clusters demonstrated previously, suggesting that this activity is a gestalt function of the GPR19-Aging nexus. This posit is reinforced by the fact that these key stress-response factors are distributed broadly across the whole PPI network, demonstrating a multifunctional and widespread role in this process.

## 4. Conclusions

Here we have demonstrated that the orphan rhodopsin-like GPCR, GPR19, has the potential, from a molecular signaling point of view, to exert a profound role in the aging process. Aging represents perhaps one of the most complex molecular programs in physiology. This pathological process encompasses the subtle interaction of a variety of physiological systems, including the insulinotropic system, immune system, management of body weight and adiposity, circadian rhythm control and stress response networks. Investigating the current state of both curated and non-curated data pertaining to GPR19, we have uncovered multiple instances in which the regulation of this receptor may facilitate a network-based trophic action over the somatic aging process. The future therapeutic investigation of this receptor, therefore, may yield important new remedial agents to treat not only specific aspects of aging (e.g., increased adiposity) but also specific forms of molecular damage (e.g., DNA damage). The therapeutic-based control of these signaling systems may therefore be able to attenuate the degree of disease progression of a wide range of disorders that are commonly linked by the fundamental pathologies found across multiple aging paradigms.

## Figures and Tables

**Figure 1 ijms-23-13598-f001:**
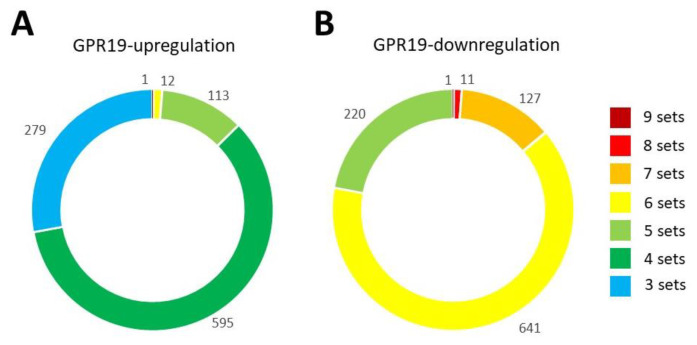
Distribution of common proteins across GEO datasets demonstrating diverse polarity of GPR19 expression alteration. (**A**) Seven divergent GEO GSE datasets (possessing a consistent upregulation of GPR19) were used to assemble a data cohort of GPR19-associated proteins. The number of datasets in which the same protein was observed is indicated by the associated color code system. (**B**) Nine divergent GEO GSE datasets (possessing a consistent downregulation of GPR19) were used to assemble a data cohort of GPR19-associated proteins. The number of datasets in which the same protein was observed is indicated by the associated color code system.

**Figure 2 ijms-23-13598-f002:**
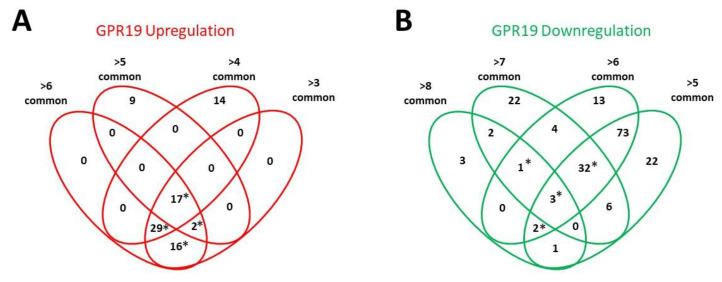
WikiPathway signaling analysis of GPR19-associated proteins. Using protein lists assembled from the GEO GPR19 upregulated cohort (proteins used were from the common to >3, >4, >5 and >6 sub-lists derived from the primary data list) or the GPR19 downregulated cohort (proteins used were from the common to >5, >6, >7 and >8 sub-lists derived from the primary data list) were subjected to WikiPathways signaling pathway enrichment analysis. For a pathway to be significantly enriched, an enrichment probability of <0.05 was required using at least two independent proteins. Venn diagrams of the enriched pathways derived from the differing GPR19 upregulated (**A**) or GPR19 downregulated (**B**) proteins were created. Pathways found in at least three out of the four (indicated by an asterisk, *) different initial datasets were then used for further analyses of their quantitative and qualitative enrichment profiles.

**Figure 3 ijms-23-13598-f003:**
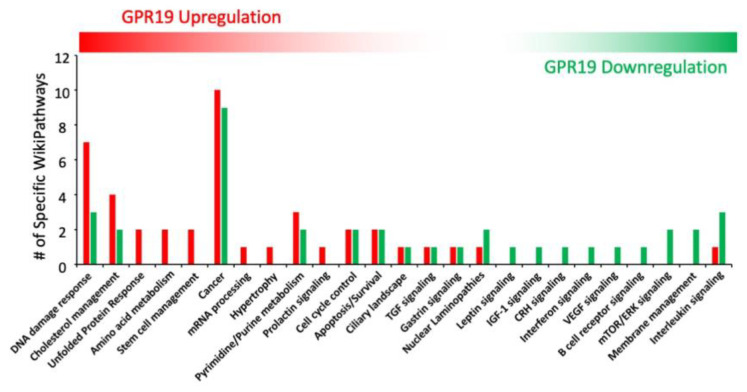
Distribution of WikiPathway significant population profile between either GPR19 upregulated or GPR19 downregulated profiles. The WikiPathways significantly enriched by at least three out of the four initial data inputs (Figure 2) were then clustered according to their divergent number of functionally similar pathways. WikiPathway groups more populated by the GPR19 upregulated datasets are left-most in the histogram, and the WikiPathway groups more populated by the GPR19 downregulated datasets are right-most in the histogram. Thus, GPR19 upregulation profiles are most strongly associated with DNA damage response, cholesterol metabolism, and the unfolded protein response, while GPR19 downregulation profiles are more strongly linked to interleukin signaling, plasma membrane regulation and mTOR signaling.

**Figure 4 ijms-23-13598-f004:**
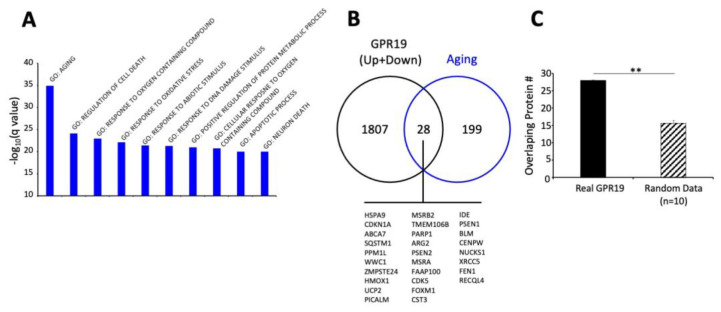
Aging-specific functionalities of GPR19-associated datasets. (**A**) Informatic benchmark analysis of an unbiased aging-specific dataset created using natural language processing. This specific resultant dataset (227 proteins) was then subject to MSigDB Gene Ontology (GO)—biological process enrichment analysis. GO annotations were only considered significant if populated by at least two independent proteins at an enrichment probability of <0.05. The most significantly populated GO term generated using this aging-specific dataset was GO: AGING (GO:0007568). (**B**) When comparing the protein contents of the specific aging dataset with the total GPR19 up plus down associated protein cohorts (Figure 1), 28 common proteins were found between these two data corpi. (**C**) This numerical level of association was highly significant as the numerical commonality between a random data set the same size as the aging dataset, and the total GPR19-associated dataset was only 15.7 + 0.81 (mean + SEM). ** indicates *p* < 0.01.

**Figure 5 ijms-23-13598-f005:**
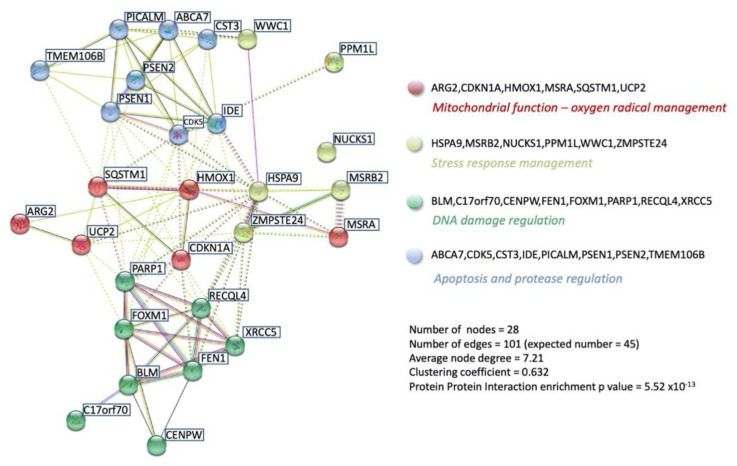
STRING network analysis of GPR19-Aging specific intersection data cohort. The STRING protein association database was used, along with k-means clustering, to assemble the 28 input proteins of the GPR19 total dataset with the aging-specific dataset (Figure 4), to create a functional classification of these 28 proteins as a coherent group. The k means clusters were color-coded by the STRING protein-protein-interaction (PPI) algorithms. The reported PPI enrichment probability of this 28-protein cohort was *p* = 5.52 × 10^−13^.

**Figure 6 ijms-23-13598-f006:**
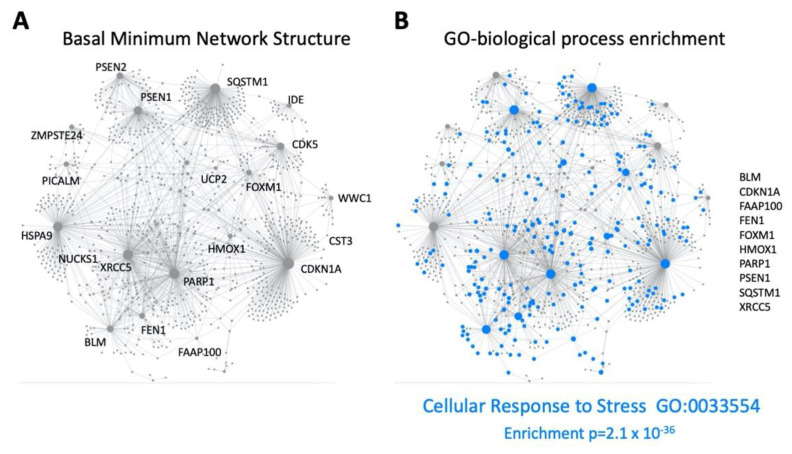
Protein-Protein-Interaction network enrichment. Using the GPR19-Aging specific intersection cohort of 28 proteins, a Basal Minimum interaction network of these 28 proteins, with bridging factors introduced using the IMEx Consortium database, was created using NetworkAnalyst (**A**). Applying Gene Ontology biological process term enrichment to this network, a strong enrichment (*p* = 2.1 × 10^−36^) was found for the GO term group, GO:0033554 ‘Cellular Response to Stress’. The proteins determining this GO term enrichment (along with their IMEX-associated proteins) in the original network are highlighted in blue and indicated to the right of the annotated network (**B**).

## Data Availability

Not applicable.

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
