# Peer review of "Intersection of the Orphan G Protein-Coupled Receptor, GPR19, with the Aging Process"

_ijms, 2022, doi:10.3390/ijms232113598_

Round 1

Reviewer 1 Report

In this review, Maudsley et al use published papers and bioinformatic approaches to link the orphan GPCR GPR19 to the aging process. They discuss what is known about GPR19 function and its potential role as a receptor for the peptide adropin, in the context of aging processes.

Comments:

The manuscript is a hybrid, with both review of the literature and apparently new analysis of existing data to derive new conclusions. Overall, the manuscript is well-written and informative. However, there are multiple instances in the text that need clarification/expansion.

1. On page 3, lines 118-132, multiple references are mis-formatted (the references have a number but also list the same reference in author/date style).

2. Some statements should cite a reference in support. These include:

-Page 3 line 134, “As we have previously discussed []” (although this might be intended to read “As discussed above”?)

-Page 4 line 189 “multidimensional signaling pathways [].”

3. Some sections make a vague statement referring to some other study without explanation of why it is relevant, or make statements that are insufficiently described; these require some additional detailed description. These include:

-Page 4, line 175, “holds tremendous promise for the future of GPCR-based aging trajectories [40].”  Describe how in 1-2 sentences.

-Page 5 line 251, “promotion of mesenchymal to epithelial transition..”  In cancer cell metastasis, MET would limit tumor cells from leaving the primary tumor, while EMT would promote tumor cell spreading. On the other hand, MET might improve tumor cell integration into a new metastatic niche. Please clarify what GPR19 does in breast cancer metastasis.

-Page 6 line 285, “A further link to aging … adaptogens.” does not actually describe why this is relevant; please expand.

4. Some sentences/sections are clumsily written or confusing and need to be rewritten more clearly. These include:

-Page 4 paragraph starting at line 191. Specifically, “linked to inhibitory cyclase” is nonsense; “linked to inhibition of (some isoforms of) adenylyl cyclase”? Line 193, “..detecting the cAMP or Ca” would be better stated like “detecting elevated levels of second messengers such as cAMP or Ca”. Line 194, “and the ligand adropin” comes before the discussion of adropin in the next paragraph, and should at least add a disclaimer (putative, proposed?), but better would be to move the next paragraph before this one. Line 197, “constitutive activity was studied in GPR19, and other orphan receptors” is both vague and overbroad. Vague because t seems only cAMP was assessed, which should be stated up front. Overbroad, unless that report studied all orphan receptors (“and some other orphan receptors”?), which it did not, and some orphans DO give constitutive activity, which is stated in the first sentence of this paragraph. This is confused and all needs clarification.

5. minor grammatical errors/typos:

-Page 3 line 139, capitalized “I” in “in vivo”

-Page 6 line 285, delete second “is”

-Page 6 line 299, “aberrates” is a very unusual word choice, perhaps “disrupts”?

-Page 8 line 355, last sentence is missing a final verb. “..and aging related dementia [] were observed.”?

6. missing information.

-Please state somewhere (paragraph page 5 line 205?) how similar GPR19 is to other receptors, and specifically the closest known deorphanized GPCRs, as this helps provide context within the GPCR family.

7. Figures

-Figure 3. The name labels should all be below the axis for a consistent look

-Figure 4A,B. The text size (especially in A) is far too small, and is pixelated if the image is expanded. Increase font size and ensure appropriate resolution.

-Figure 5. Gene name labels are too small, and also pixelated. Increase font size and ensure appropriate resolution.

-Figure 6B. Gene name labels are too small, and also pixelated, and in a non-black font which further reduces readability of the gene name text. Increase font size, change to black (or make extra large to compensate) and ensure appropriate resolution. Also consider changing the order to alphabetical, as the top-to-bottom order from panel A is not particularly meaningful in B (or label the relevant nodes on the graph itself as in A).

Reviewer 2 Report

I really enjoyed reading the manuscript. It is full of good data that introduce a reader into the role of GPR19 and by both significant amount of experimentally data and bioinformatics methodology show its role in the aging process. The review suggests the future therapeutic investigation of this receptor as very promising. It is a real comprehensive review that comes from respected authors that did a lot of experimentation and provided many impressive data in this multidisciplinary field. This review is a significant contribution to the knowledge of GPCRs and, especially GPR19.

I have just a few suggestion.

Abstract tells about GPCRs, but the review is focused on GPR19. It will be much useful for readers to abstract point just toward GPR19.  

Part 1.1. second paragraph do not provide direct and clear connection to aging, but last two sentence are well connection to the last one in the first paragraph. I suggest to move last two sentence of the second paragraph to the end of the first, and the rest of the second to introduction.

Figures have to be larger to be better visible and legible. I do not know if this is on editorial or authors.

Reviewer 3 Report

The author tried to document the link between GPR19 and aging process.  I think this is poor description. From the author's description, I did not see the inevitable connection of GPR19 and age -related diseases.

I have the following recommendations:

1.In Introduction section, Author gave a big picture of general knowledge of GPCR. However, I did not see any information related to GPR19 and aging process.

2. The GPR19 and aging-related activity section is disorganized.  

A better description method should be from the gene cloning, tissue expression, functional research (Knockdown or knockout model, activation or inhibition), ligands, and intracellular signaling. Further explain association of GPR19 or its ligands and aging process.

3. Several important publications on GPR19-mediated aging process are missing and should be cited (for example PMID: 34462439 and PMID: 29896421).

4. For abbreviation, the correct approach should be the first time there should be full name and abbreviation, then directly use abbreviation, should not repeatedly appear full name and abbreviation. Please check the full manuscript.

5. Line 227-228: This sentence also should be re-written.

6. Line 118-132: Incorrect reference style

Round 2

Reviewer 3 Report

The authors have included the corrections, so it can be accepted for publication